# INCREMENTAL LEARNING OF STRUCTURED MEMORY VIA CLOSED-LOOP TRANSCRIPTION

**Shengbang Tong**[1]**, Xili Dai**[2]**, Ziyang Wu**[1]**, Mingyang Li**[3]**, Brent Yi**[1]**, Yi Ma**[1, 3]

[1] University of California, Berkeley, [2] The Hong Kong University of Science and Technology(Guangzhou)
[3] Tsinghua-Berkeley Shenzhen Institute (TBSI), Tsinghua University

## ABSTRACT

This work proposes a minimal computational model for learning structured memories of multiple object classes in an incremental setting. Our approach is based on establishing a *closed-loop transcription* between the classes and a corresponding set of subspaces, known as a linear discriminative representation, in a low-dimensional feature space. Our method is simpler than existing approaches for incremental learning, and more efficient in terms of model size, storage, and computation: it requires only a single, fixed-capacity autoencoding network with a feature space that is used for both discriminative and generative purposes. Network parameters are optimized simultaneously without architectural manipulations, by solving a constrained minimax game between the encoding and decoding maps over a single rate reduction-based objective. Experimental results show that our method can effectively alleviate catastrophic forgetting, achieving significantly better performance than prior work of generative replay on MNIST, CIFAR-10, and ImageNet-50, despite requiring fewer resources.

## 1 INTRODUCTION

Artificial neural networks have demonstrated a great ability to learn representations for hundreds or even thousands of classes of objects, in both discriminative and generative contexts. However, networks typically must be trained offline, with uniformly sampled data from all classes simultaneously. When the same network is updated to learn new classes without data from the old ones, previously learned knowledge will fall victim to the problem of *catastrophic forgetting* (McCloskey & Cohen, 1989). This is known in neuroscience as the stability-plasticity dilemma: the challenge of ensuring that a neural system can learn from a new environment while retaining essential knowledge from previous ones (Grossberg, 1987).

In contrast, natural neural systems (e.g. animal brains) do not seem to suffer from such catastrophic forgetting at all. They are capable of developing new memory of new objects while retaining memory of previously learned objects. This ability, for either natural or artificial neural systems, is often referred to as *incremental learning, continual learning, sequential learning*, or *life-long learning* (Allred & Roy, 2020).

While many recent works have highlighted how incremental learning might enable artificial neural systems that are trained in more flexible ways, the strongest existing efforts toward answering the stability-plasticity dilemma for artificial neural networks typically require raw exemplars (Rebuffi et al., 2017; Chaudhry et al., 2019b) or require external task information (Kirkpatrick et al., 2017). Raw exemplars, particularly in the case of high-dimensional inputs like images, are costly and difficult to scale, while external mechanisms — which, as surveyed in Section 2, include secondary networks and representation spaces for generative replay, incremental allocation of network resources, network duplication, or explicit isolation of used and unused parts of the network — require heuristics and incur hidden costs.

In this work, we are interested in an incremental learning setting that counters these trends with two key qualities. (1) The first is that it is *memory-based*. When learning new classes, no raw exemplars of old classes are available to train the network together with new data. This implies that one has to rely on a compact and thus structured "memory" of old classes, such as incrementally learned generative representations of the old classes, as well as the associated encoding and decoding mappings (Kemker

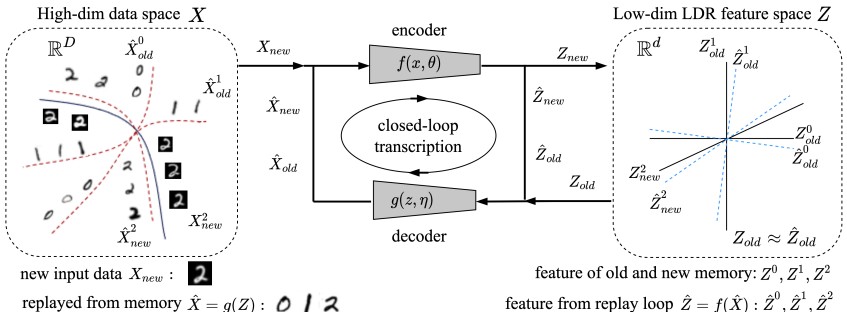

Figure 1: **Overall framework** of our closed-loop transcription based incremental learning for a structured LDR memory. Only a single, entirely self-contained, encoding-decoding network is needed: for a new data class $\boldsymbol{X}_{new}$, a new LDR memory $\boldsymbol{Z}_{new}$ is incrementally learned as a minimax game between the encoder and decoder subject to the constraint that old memory of past classes $\boldsymbol{Z}_{old}$ is intact through the closed-loop transcription (or replay): $\boldsymbol{Z}_{old} \approx \hat{\boldsymbol{Z}}_{old} = f(g(\boldsymbol{Z}_{old}))$.

& Kanan, 2018). (2) The second is that it is *self-contained.* Incremental learning takes place in a single neural system with a fixed capacity, and in a common representation space. The ability to minimize forgetting is implied by optimizing an overall learning objective, without external networks, architectural modifications, or resource allocation mechanisms.

Concretely, the contributions of our work are as follows:

**(1)** We demonstrate how the *closed-loop transcription* (CTRL) framework (Dai et al., 2022; 2023) can be adapted for memory-based, self-contained mitigation of catastrophic forgetting (Figure 1). To the best of our knowledge, these qualities have not yet been demonstrated by existing methods. Closed-loop transcription aims to learn linear discriminative representations (LDRs) via a rate reduction-based (Yu et al., 2020; Ma et al., 2007; Ding et al., 2023) minimax game: our method, which we call incremental closed-loop transcription (i-CTRL), shows how these principled representations and objectives can uniquely facilitate incremental learning of stable and structured class memories. This requires only a fixed-sized neural system and a common learning objective, which transforms the standard CTRL minimax game into a constrained one, where the goal is to optimize a rate reduction objective for each new class while keeping the memory of old classes intact.

**(2)** We quantitatively evaluate i-CTRL on class-incremental learning for a range of datasets: MNIST (LeCun et al., 1998), CIFAR-10 (Krizhevsky et al., 2009), and ImageNet-50 (Deng et al., 2009). Despite requiring fewer resources (smaller network and nearly no extra memory buffer), i-CTRL outperforms comparable alternatives: it achieves a 5.8% improvement in average classification accuracy over the previous state of the art on CIFAR-10, and a 10.6% improvement in average accuracy on ImageNet-50.

**(3)** We qualitatively verify the structure and generative abilities of learned representations. Notably, the self-contained i-CTRL system's common representation is used for both classification and generation, which eliminates the redundancy of external generative replay representations used by prior work.

**(4)** We demonstrate a "class-unsupervised" incremental reviewing process for i-CTRL. As an incremental neural system learns more classes, the memory of previously learned classes inevitably degrades: by seeing a class only once, we can only expect to form a temporary memory. Facilitated by the structure of our linear discriminative representations, the incremental reviewing process shows that the standard i-CTRL objective function can reverse forgetting in a trained i-CTRL system using samples from previously seen classes *even if they are unlabeled.* The resulting semi-supervised process improves generative quality and raises the accuracy of i-CTRL from 59.9% to 65.8% on CIFAR-10, achieving jointly-optimal performance despite only incrementally provided class labels.

## 2 RELATED WORK

A significant body of work has studied methods for addressing forms of the incremental learning problem. In this section, we discuss a selection of representative approaches, and highlight relationships to i-CTRL.

In terms of *how new data classes are provided and tested*, incremental learning methods in the literature can be roughly divided into two groups. The first group addresses *task* incremental learning (task-IL), where a model is sequentially trained on multiple tasks where each task may contain multiple classes to learn. At test time, the system is asked to classify data seen so far, *provided with a task identifier indicating which task the test data is drawn from*. The second group, which many recent methods fall under, tackles *class*-incremental learning (class-IL). Class-IL is similar to task-IL but does not require a task identitifier at inference. Class-IL is therefore more challenging, and is the setting considered by this work.

In terms of *what information incremental learning relies on*, existing methods mainly fall into the following categories.

**Regularization-based** methods introduce penalty terms designed to mitigate forgetting of previously trained tasks. For instance, Elastic Weight Consolidation (EWC) (Kirkpatrick et al., 2017) and Synaptic Intelligence (SI) (Zenke et al., 2017) limit changes of model parameters deemed to be important for previous tasks by imposing a surrogate loss. Alternatively, Learning without Forgetting (LwF) (Li & Hoiem, 2017) utilizes a knowledge distillation loss to prevent large drifts of the model weights during training on the current task. Although these methods, which all apply regularization on network parameters, have demonstrated competitive performance on task-IL scenarios, our evaluations (Table 1) show that their performance does not transfer to the more challenging class-IL settings.

**Architecture-based** methods explicitly alter the network architecture to incorporate new classes of data. Methods such as Dynamically Expandable Networks (DEN) (Yoon et al., 2017), Progressive Neural Networks (PNN) (Rusu et al., 2016), Dynamically Expandable Representation (DER) (Yan et al., 2021) and ReduNet (Wu et al., 2021) add new neural modules to the existing network when required to learn a new task. Since these methods are not dealing with a self-contained network with a fixed capacity, one disadvantage of these methods is therefore their memory footprint: their model size often grows linearly with the number of tasks or classes. Most architecture-based methods target the less challenging task-IL problems and are not suited for class-IL settings. In contrast, our work addresses the class-IL setting with only a simple, off-the-shelf network (see Appendix B for details). Note that the method Redunet also uses rate-reduction inspired objective function to conduct class incremental learning. Our method is different from the method that (i) our method does not require dynamic expansion of the network (ii) Our method aims to learn a continuous encoder and decoder.(iii) Empirically, our method has shown better performance and scalability.

**Exemplar-based** methods combat forgetting by explicitly retaining data from previously learned tasks. Most early memory-based methods, such as iCaRL (Rebuffi et al., 2017) and ER (Chaudhry et al., 2019a), store a subset of raw data samples from each learned class, which is used along with the new classes to jointly update the model. A-GEM (Chaudhry et al., 2018) also relies on storing such an exemplar set: rather than directly training with new data, A-GEM calculates a reference gradient from the stored data and projects the gradient from the new task onto these reference directions in hope of maintaining performance on old tasks. While these methods have demonstrated promising results, storing raw data of learned classes is unnatural from a neuroscientific perspective(Robins, 1995) and resource-intensive, particularly for higher-dimensional inputs. A fair comparison is thus *not* possible: the structured memory used by i-CTRL is highly compact in comparison, but as demonstrated in Section 4 still outperforms several exemplar-based methods.

**Generative memory-based** methods use generative models such as GANs or autoencoders for replaying data for old tasks or classes, rather than storing raw samples and exemplars. Methods such as Deep Generative Replay (DGR) (Shin et al., 2017), Memory Replay Gans (MeRGAN) (Wu et al., 2018), and Dynamic Generative Memory (DGM) (Ostapenko et al., 2019) propose to train a GAN on previously seen classes and use synthesized data to alleviate forgetting when training on new tasks. Methods like DAE (Zhou et al., 2012) learn with add and merge feature strategy. To further improve memory efficiency, methods such as FearNet (Kemker & Kanan, 2018) and EEC (Ayub & Wagner, 2020) store intermediate features of old classes and use these more compact representations for generative replay. Existing generative memory-based approaches have performed competitively on class-IL without storing raw data samples, but require separate networks and feature representations for generative and discriminative purposes. Our comparisons are primarily focused on this line of work, as our approach also uses a generative memory for incremental learning. Uniquely, however, we do so with only a single closed-loop encoding-decoding network and store a minimum amount of

information – a mean and covariance – for each class. This closed-loop generative model is more stable to train (Dai et al., 2022; Tong et al., 2022), and improves resource efficiency by obviating the need to train separate generative and discriminative representations.

# 3 METHOD

## 3.1 LINEAR DISCRIMINATIVE REPRESENTATION AS MEMORY

Consider the task of learning to memorize $k$ classes of objects from images. Without loss of generality, we may assume that images of each class belong to a low-dimensional submanifold in the space of images $\mathbb{R}^D$, denoted as $\mathcal{M}_j$, for $j = 1, \ldots, k$. Typically, we are given $n$ samples $\boldsymbol{X} = [\boldsymbol{x}^1, \ldots, \boldsymbol{x}^n] \subset \mathbb{R}^{D \times n}$ that are partitioned into $k$ subsets $\boldsymbol{X} = \cup_{j=1}^k \boldsymbol{X}_j$, with each subset $\boldsymbol{X}_j$ sampled from $\mathcal{M}_j, j = 1, \ldots, k$. The goal here is to learn a compact representation, or a "memory", of these $k$ classes from these samples, which can be used for both discriminative (e.g. classification) and generative purposes (e.g. sampling and replay).

**Autoencoding.** We model such a memory with an *autoencoding* tuple $\{f, g, \boldsymbol{z}\}$ that consists of an encoder $f(\cdot, \theta)$ parameterized by $\theta$, that maps the data $\boldsymbol{x} \in \mathbb{R}^D$ continuously to a compact feature $\boldsymbol{z}$ in a much lower-dimensional space $\mathbb{R}^d$, and a decoder $g(\cdot, \eta)$ parameterized by $\eta$, that maps a feature $\boldsymbol{z}$ back to the original data space $\mathbb{R}^D$:

$$f(\cdot, \theta) : \boldsymbol{x} \mapsto \boldsymbol{z} \in \mathbb{R}^d; \quad g(\cdot, \eta) : \boldsymbol{z} \mapsto \hat{\boldsymbol{x}} \in \mathbb{R}^D. \tag{1}$$

For the set of samples $\boldsymbol{X}$, we let $\boldsymbol{Z} = f(\boldsymbol{X}, \theta) \doteq [\boldsymbol{z}^1, \ldots, \boldsymbol{z}^n] \subset \mathbb{R}^{d \times n}$ with $\boldsymbol{z}^i = f(\boldsymbol{x}^i, \theta) \in \mathbb{R}^d$ be the set of corresponding features. Similarly let $\hat{\boldsymbol{X}} \doteq g(\boldsymbol{Z}, \eta)$ be the decoded data from the features. The autoencoding tuple can be illustrated by the following diagram:

$$\boldsymbol{X} \xrightarrow{f(\boldsymbol{x}, \theta)} \boldsymbol{Z} \xrightarrow{g(\boldsymbol{z}, \eta)} \hat{\boldsymbol{X}}. \tag{2}$$

We refer to such a learned tuple: $\{f(\cdot, \theta), g(\cdot, \eta), \boldsymbol{Z}\}$ as a compact "memory" for the given dataset $\boldsymbol{X}$.

**Structured LDR autoencoding.** For such a memory to be convenient to use for subsequent tasks, including incremental learning, we would like a representation $\boldsymbol{Z}$ that has well-understood structures and properties. Recently, Chan *et al.* (Chan et al., 2021) proposed that for both discriminative and generative purposes, $\boldsymbol{Z}$ should be a *linear discriminative representation* (LDR). More precisely, let $\boldsymbol{Z}_j = f(\boldsymbol{X}_j, \theta), j = 1, \ldots, k$ be the set of features associated with each of the $k$ classes. Then each $\boldsymbol{Z}_j$ should lie on a low-dimensional linear subspace $\mathcal{S}_j$ in $\mathbb{R}^d$ which is highly incoherent (ideally orthogonal) to others $\mathcal{S}_i$ for $i \neq j$. Notice that the linear subspace structure enables both interpolation and extrapolation, and incoherence between subspaces makes the features discriminative for different classes. As we will see, these structures are also easy to preserve when incrementally learning new classes.

## 3.2 LEARNING LDR VIA CLOSED-LOOP TRANSCRIPTION

As shown in (Yu et al., 2020), the incoherence of learned LDR features $\boldsymbol{Z} = f(\boldsymbol{X}, \theta)$ can be promoted by maximizing a coding rate reduction objective, known as *the MCR$^2$ principle*:

$$\max_\theta \ \Delta R(\boldsymbol{Z}) = \Delta R(\boldsymbol{Z}_1, \ldots, \boldsymbol{Z}_k) \doteq \underbrace{\frac{1}{2} \log \det \left( \boldsymbol{I} + \alpha \boldsymbol{Z} \boldsymbol{Z}^* \right)}_{R(\boldsymbol{Z})} - \sum_{j=1}^k \gamma_j \underbrace{\frac{1}{2} \log \det \left( \boldsymbol{I} + \alpha_j \boldsymbol{Z}_j \boldsymbol{Z}_j^* \right)}_{R(\boldsymbol{Z}_j)},$$

where, for a prescribed quantization error $\epsilon$, $\alpha = \frac{d}{n \epsilon^2}$, $\alpha_j = \frac{d}{|\boldsymbol{Z}_j| \epsilon^2}$, $\gamma_j = \frac{|\boldsymbol{Z}_j|}{n}$.

As noted in (Yu et al., 2020), maximizing the rate reduction promotes learned features that span the entire feature space. It is therefore not suitable to naively apply for the case of incremental learning, as the number of classes increases within a fixed feature space.[1] The closed-loop transcription (CTRL) framework introduced by (Dai et al., 2022) suggests resolving this challenge by learning the encoder $f(\cdot, \theta)$ and decoder $g(\cdot, \eta)$ together as a minimax game: while the encoder tries to maximize the rate reduction objective, the decoder should minimize it instead. That is, the decoder $g$ minimizes

---

[1] As the number of classes is initially small in the incremental setting, if the dimension of the feature space $d$ is high, maximizing the rate reduction may over-estimate the dimension of each class.

resources (measured by the coding rate) needed for the replayed data for each class $\hat{\boldsymbol{X}}_j = g(\boldsymbol{Z}_j, \eta)$, decoded from the learned features $\boldsymbol{Z}_j = f(\boldsymbol{X}_j, \theta)$, to emulate the original data $\boldsymbol{X}_j$ well enough. As it is typically difficult to directly measure the similarity between $\boldsymbol{X}_j$ and $\hat{\boldsymbol{X}}_j$, (Dai et al., 2022) proposes measuring this similarity with the *rate reduction* of their corresponding features $\boldsymbol{Z}_j$ and $\hat{\boldsymbol{Z}}_j = f(\hat{\boldsymbol{X}}_j(\theta, \eta), \theta)(\cup$ here represents concatenation):

$$\Delta R(\boldsymbol{Z}_j, \hat{\boldsymbol{Z}}_j) \doteq R(\boldsymbol{Z}_j \cup \hat{\boldsymbol{Z}}_j) - \frac{1}{2}\big(R(\boldsymbol{Z}_j) + R(\hat{\boldsymbol{Z}}_j)\big). \tag{3}$$

The resulting $\Delta R$ gives a principled "distance" between subspace-like Gaussian ensembles, with the property that $\Delta R(\boldsymbol{Z}_j, \hat{\boldsymbol{Z}}_j) = 0$ iff $\mathrm{Cov}(\boldsymbol{Z}_j) = \mathrm{Cov}(\hat{\boldsymbol{Z}}_j)$ (Ma et al., 2007).

$$\min_\theta \max_\eta \Delta R(\boldsymbol{Z}) + \Delta R(\hat{\boldsymbol{Z}}) + \sum_{j=1}^{k} \Delta R(\boldsymbol{Z}_j, \hat{\boldsymbol{Z}}_j), \tag{4}$$

one can learn a good LDR $\boldsymbol{Z}$ when optimized jointly for all $k$ classes. The learned representation $\boldsymbol{Z}$ has clear incoherent linear subspace structures in the feature space which makes them very convenient to use for subsequent tasks (both discriminative and generative).

### 3.3 INCREMENTAL LEARNING WITH AN LDR MEMORY

The incoherent linear structures for features of different classes closely resemble how objects are encoded in different areas of the inferotemporal cortex of animal brains (Chang & Tsao, 2017; Bao et al., 2020). The closed-loop transcription $\boldsymbol{X} \to \boldsymbol{Z} \to \hat{\boldsymbol{X}} \to \hat{\boldsymbol{Z}}$ also resembles popularly hypothesized mechanisms for memory formation (Ven et al., 2020; Josselyn & Tonegawa, 2020). This leads to a question: since memory in the brains is formed in an incremental fashion, can the above closed-loop transcription framework also support incremental learning?

**LDR memory sampling and replay.** The simple linear *structures* of LDR make it uniquely suited for incremental learning: the distribution of features $\boldsymbol{Z}_j$ of each previously learned class can be explicitly and concisely represented by a principal subspace $\mathcal{S}_j$ in the feature space. To preserve the memory of an old class $j$, we only need to preserve the subspace while learning new classes. To this end, we simply sample $m$ representative prototype features on the subspace along its top $r$ principal components, and denote these features as $\boldsymbol{Z}_{j,old}$. Because of the simple linear structures of LDR, we can sample from $\boldsymbol{Z}_{j,old}$ by calculating the mean and covariance of $\boldsymbol{Z}_{j,old}$ after learning class $j$. The storage required is extremely small, since we only need to store means and covariances, which are sampled from as needed. Suppose a total of $t$ old classes have been learned so far. If prototype features, denoted $\boldsymbol{Z}_{old} \doteq [\boldsymbol{Z}_{old}^1, \ldots, \boldsymbol{Z}_{old}^t]$, for all of these classes can be preserved when learning new classes, the subspaces $\{\mathcal{S}_j\}_{j=1}^t$ representing past memory will be preserved as well. Details about sampling and calculating mean and convariance can be found in the Appendix 1 and Appendix 2

**Incremental learning LDR with an old-memory constraint.** Notice that, with the learned auto-encoding (2), one can replay and use the images, say $\hat{\boldsymbol{X}}_{old} = g(\boldsymbol{Z}_{old}, \eta)$, associated with the memory features to avoid forgetting while learning new classes. This is typically how generative models have been used for prior incremental learning methods. However, with the closed-loop framework, explicitly replaying images from the features is not necessary. Past memory can be effectively preserved through optimization exclusively on the features themselves.

Consider the task of incrementally learning a new class of objects.[2] We denote a corresponding new sample set as $\boldsymbol{X}_{new}$. The features of $\boldsymbol{X}_{new}$ are denoted as $\boldsymbol{Z}_{new}(\theta) = f(\boldsymbol{X}_{new}, \theta)$. We concatenate them together with the prototype features of the old classes $\boldsymbol{Z}_{old}$ and form $\boldsymbol{Z} = [\boldsymbol{Z}_{new}(\theta), \boldsymbol{Z}_{old}]$. We denote the replayed images from all features as $\hat{\boldsymbol{X}} = [\hat{\boldsymbol{X}}_{new}(\theta, \eta), \hat{\boldsymbol{X}}_{old}(\eta)]$ although we do not actually need to compute or use them explicitly. We only need features of replayed images, denoted $\hat{\boldsymbol{Z}} = f(\hat{\boldsymbol{X}}, \theta) = [\hat{\boldsymbol{Z}}_{new}(\theta, \eta), \hat{\boldsymbol{Z}}_{old}(\theta, \eta)]$.

Mirroring the motivation for the multi-class CTRL objective (4), we would like the features of the new class $\boldsymbol{Z}_{new}$ to be incoherent to all of the old ones $\boldsymbol{Z}_{old}$. As $\boldsymbol{Z}_{new}$ is the only new class whose features needs to be learned, the objective (4) reduces to the case where $k = 1$:

---

[2] In Appendix A, we consider the more general setting where the task contains a small batch of new classes, and present algorthmic details in that general setting.

$$\min_{\eta} \max_{\theta} \Delta R(\boldsymbol{Z}) + \Delta R(\hat{\boldsymbol{Z}}) + \Delta R(\boldsymbol{Z}_{new}, \hat{\boldsymbol{Z}}_{new}). \tag{5}$$

However, when we update the network parameters $(\theta, \eta)$ to optimize the features for the new class, the updated mappings $f$ and $g$ will change features of the old classes too. Hence, to minimize the distortion of the old class representations, we can try to enforce $\mathrm{Cov}(\boldsymbol{Z}_{j,old}) = \mathrm{Cov}(\hat{\boldsymbol{Z}}_{j,old})$. In other words, while learning new classes, we enforce the memory of old classes remain "self-consistent" through the transcription loop:

$$\boldsymbol{Z}_{old} \xrightarrow{g(\boldsymbol{z}, \eta)} \hat{\boldsymbol{X}}_{old} \xrightarrow{f(\boldsymbol{x}, \theta)} \hat{\boldsymbol{Z}}_{old}. \tag{6}$$

Mathematically, this is equivalent to setting $\Delta R(\boldsymbol{Z}_{old}, \hat{\boldsymbol{Z}}_{old}) \doteq \sum_{j=1}^{t} \Delta R(\boldsymbol{Z}_{j,old}, \hat{\boldsymbol{Z}}_{j,old}) = 0$. Hence, the above minimax program (5) is revised as a *constrained* minimax game, which we refer to as *incremental closed-loop transcription* (i-CTRL). The objective of this game is identical to the standard multi-class CTRL objective (4), but includes just one additional constraint:

$$\min_{\eta} \max_{\theta} \quad \Delta R(\boldsymbol{Z}) + \Delta R(\hat{\boldsymbol{Z}}) + \Delta R(\boldsymbol{Z}_{new}, \hat{\boldsymbol{Z}}_{new})$$

$$\text{subject to} \quad \Delta R(\boldsymbol{Z}_{old}, \hat{\boldsymbol{Z}}_{old}) = 0. \tag{7}$$

In practice, the constrained minimax program can be solved by *alternating* minimization and maximization between the encoder $f(\cdot, \theta)$ and decoder $g(\cdot, \eta)$ as follows:

$$\max_{\theta} \quad \Delta R(\boldsymbol{Z}) + \Delta R(\hat{\boldsymbol{Z}}) + \lambda \cdot \Delta R(\boldsymbol{Z}_{new}, \hat{\boldsymbol{Z}}_{new}) - \gamma \cdot \Delta R(\boldsymbol{Z}_{old}, \hat{\boldsymbol{Z}}_{old}), \tag{8}$$

$$\min_{\eta} \quad \Delta R(\boldsymbol{Z}) + \Delta R(\hat{\boldsymbol{Z}}) + \lambda \cdot \Delta R(\boldsymbol{Z}_{new}, \hat{\boldsymbol{Z}}_{new}) + \gamma \cdot \Delta R(\boldsymbol{Z}_{old}, \hat{\boldsymbol{Z}}_{old}); \tag{9}$$

where the constraint $\Delta R(\boldsymbol{Z}_{old}, \hat{\boldsymbol{Z}}_{old}) = 0$ in (7) has been converted (and relaxed) to a Lagrangian term with a corresponding coefficient $\gamma$ and sign. We additionally introduce another coefficient $\lambda$ for weighting the rate reduction term associated with the new data. More algorithmic details are given in Appendix A.

**Jointly optimal memory via incremental reviewing.** As we will see, the above constrained minimax program can already achieve state of the art performance for incremental learning. Nevertheless, developing an optimal memory for *all classes* cannot rely on graceful forgetting alone. Even for humans, if an object class is learned only once, we should expect the learned memory to fade as we continue to learn new others, unless the memory can be consolidated by reviewing old object classes.

To emulate this phase of memory forming, after incrementally learning a whole dataset, we may go back to review all classes again, one class at a time. We refer to going through all classes once as one reviewing "cycle".[3] If needed, multiple reviewing cycles can be conducted. It is quite expected that reviewing can improve the learned (LDR) memory. But somewhat surprisingly, the closed-loop framework allows us to review even in a "class-unsupervised" manner: when reviewing data of an old class say $\boldsymbol{X}_j$, the system does not need the class label and can simply treat $\boldsymbol{X}_j$ as a new class $\boldsymbol{X}_{new}$. That is, the system optimizes the same constrained mini-max program (7) without any modification; after the system is optimized, one can identify the newly learned subspace spanned by $\boldsymbol{Z}_{new}$, and use it to replace or merge with the old subspace $\mathcal{S}_j$. As our experiments show, such an class-unsupervised incremental review process can gradually improve both discriminative and generative performance of the LDR memory, eventually converging to that of a jointly-learned memory.

## 4 EXPERIMENTAL VERIFICATION

We now evaluate the performance of our method and compare with several representative incremental learning methods. Since different methods have very different requirements in data, networks, and computation, it is impossible to compare all in the same experimental conditions. For a fair comparison, we do not compare with methods that deviate significantly from the IL setting that our method is designed for: as examples, this excludes methods that rely on feature extracting networks pre-trained on additional datasets such as FearNet (Kemker & Kanan, 2018) or methods that expand the feature space such as DER (Yan et al., 2021). Instead, we demonstrate the effectiveness of our method by choosing baselines that can be trained using similar *fixed* network architectures without any pretraining. Nevertheless, most existing incremental learning methods that we can compare against still rely on a buffer that acts as a memory of past tasks. They require significantly more storage than i-CTRL, which only needs to track first and second moments of each seen class (see Appendix A for algorithm implementation details).

---

[3] to distinguish from the term "epoch" used in the conventional joint learning setting.

| Category | Method | MNIST | | | | CIFAR-10 | | | |
|---|---|---|---|---|---|---|---|---|---|
| | | 10-splits | | 5-splits | | 10-splits | | 5-splits | |
| | | Last | Avg | Last | Avg | Last | Avg | Last | Avg |
| *Regularization* | LwF (Li & Hoiem, 2017) | - | - | 0.196 | 0.455 | - | - | 0.196 | 0.440 |
| | SI (Zenke et al., 2017) | - | - | 0.193 | 0.461 | - | - | 0.196 | 0.441 |
| *Architecture* | ReduNet (Wu et al., 2021) | - | - | 0.961 | 0.982 | - | - | 0.539 | 0.645 |
| *Exemplar* | iCaRL (Rebuffi et al., 2017) | 0.322 | 0.588 | 0.725 | 0.803 | 0.212 | 0.431 | 0.487 | 0.632 |
| | A-GEM (Chaudhry et al., 2018) | 0.382 | 0.574 | 0.597 | 0.764 | 0.115 | 0.293 | 0.204 | 0.473 |
| | CLR-ER (Arani et al., 2022) | - | - | - | 0.895 | - | - | - | 0.662 |
| *Generative* | DGMw (Ostapenko et al., 2019) | - | 0.965 | - | - | - | 0.562 | - | - |
| | EEC (Ayub & Wagner, 2020) | - | 0.978 | - | - | - | 0.669 | - | - |
| | EECS (Ayub & Wagner, 2020) | - | 0.963 | - | - | - | 0.619 | - | - |
| | i-CTRL (ours) | **0.975** | **0.989** | **0.978** | **0.990** | **0.599** | **0.727** | **0.627** | **0.723** |

Table 1: Comparison on MNIST and CIFAR-10.

## 4.1 DATASETS, NETWORKS, AND SETTINGS

We conduct experiments on the following datasets: MNIST (LeCun et al., 1998), CIFAR-10 (Krizhevsky et al., 2014), and ImageNet-50 (Deng et al., 2009). All experiments are conducted for the more challenging class-IL setting. For both MNIST and CIFAR-10, the 10 classes are split into 5 tasks with 2 classes each or 10 tasks with 1 class each; for ImageNet-50, the 50 classes are split into 5 tasks of 10 classes each. For MNIST and CIFAR-10 experiments, for the encoder $f$ and decoder $g$, we adopt a very simple network architecture modified from DCGAN (Radford et al., 2016), which is merely a *four-layer* convolutional network. For ImageNet-50, we use a deeper version of DCGAN which contains only 40% of the standard ResNet-18 structure.

## 4.2 COMPARISON OF CLASSIFICATION PERFORMANCE

We first evaluate the memory learned (without review) for classification. Similar to (Yu et al., 2020), we adopt a simple *nearest subspace* algorithm for classification, with details given in Appendix B. Unlike other generative memory-based incremental learning approaches, note that we do *not* need to train a separate network for classification.

**MNIST and CIFAR-10.** Table 1 compares i-CTRL against representative SOTA generative-replay incremental learning methods in different categories on the MNIST and CIFAR-10 datasets. We report results for both 10-splits and 5-splits, in terms of both last accuracy and average accuracy (following definition in iCaRL (Rebuffi et al., 2017)). Results on regularization-based and exemplar-based methods are obtained by adopting the same benchmark and training protocol as in (Buzzega et al., 2020). All other results are based on publicly available code released by the original authors. We reproduce all exemplar-based methods with a buffer size no larger than 2000 raw images or features for MNIST and CIFAR-10, which is a conventional buffer size used in other methods. Compared to these methods, i-CTRL uses a single smaller network and only needs to store means and covariances.

For a simple dataset like MNIST, we observe that i-CTRL outperforms all current SOTA on both settings. In the 10-task scenario, it is 1% higher on average accuracy, despite the SOTA is already as high as 97.8%. In general incremental learning methods achieve better performance for smaller number of steps. Here, the 10-step version even outperforms all other methods in the 5-step setting.

For CIFAR-10, we observe more significant improvement. For incremental learning with more tasks (i.e splits = 10), to our best knowledge, EEC/EECS (Ayub & Wagner, 2020) represents the current SOTA. Despite the fact that EEC uses multiple autoencoders and requires a significantly larger amount of memory (see Table 6 in the appendix), we see that i-CTRL outperforms EEC by more than 3%. For a more fair comparison, we have also included results of EECS from the same paper, which aggregate all autoencoders into one. i-CTRL outperforms EECS by nearly 10%. We also observe that i-CTRL with 10 steps is again better than all current methods that learn with 5 steps, in terms of both last and average accuracy.

| iCaRL-S | EEIL-S | DGMw | EEC | EECS | i-CTRL |
|---|---|---|---|---|---|
| 0.290 | 0.118 | 0.178 | 0.352 | 0.309 | **0.458** |

Table 2: Comparison on ImageNet-50. The results of other methods are as reported in the EEC paper.

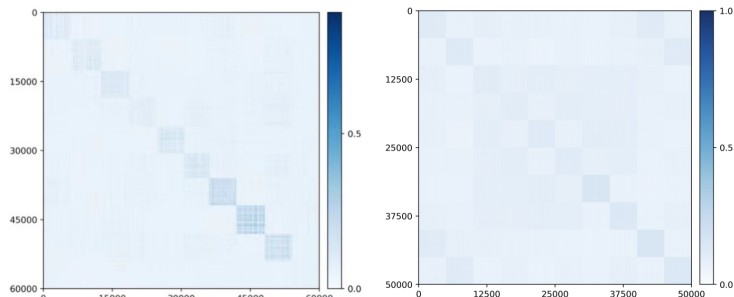

Figure 2: Block diagonal structure of $|\boldsymbol{Z}^\top \boldsymbol{Z}|$ in the feature space for MNIST (left) and CIFAR-10 (right).

**ImageNet-50.** We also evaluate and compare our method on ImageNet-50, which has a larger number of classes and higher resolution inputs. Training details can be found in Appendix B. We adopt results from (Ayub & Wagner, 2020) and report average accuracy across five splits. From the table, we observe the same trend, a very significant improvement from the previous methods by almost 10%! Since ImageNet-50 is a more complicated dataset, we can even further improve the performance using augmentation. More discussion can be found in Appendix 11.

### 4.3 GENERATIVE PROPERTIES OF THE LEARNED LDR MEMORY

Unlike some of the incremental methods above, which learn models only for classification purposes (as those in Table 1), the i-CTRL model is both discriminative and generative. In this section, we show the generative abilities of our model and visualize the structure of the learned memory. We also include standard metrics for analysis in Appendix H.

**Visualizing auto-encoding properties.** We begin by qualitatively visualizing some representative images $\boldsymbol{X}$ and the corresponding replayed $\hat{\boldsymbol{X}}$ on MNIST and CIFAR-10. The model is learned incrementally with the datasets split into 5 tasks. Results are shown in Figure 3, where we observe that the reconstructed $\hat{\boldsymbol{X}}$ preserves the main visual characteristics of $\boldsymbol{X}$ including shapes and textures. For a simpler dataset like MNIST, the replayed $\hat{\boldsymbol{X}}$ are almost identical to the input $\boldsymbol{X}$! This is rather remarkable given: (1) our method does not explicitly enforce $\hat{\boldsymbol{x}} \approx \boldsymbol{x}$ for individual samples as most autoencoding methods do, and (2) after having incrementally learned all classes, the generator has not forgotten how to generate digits learned earlier, such as 0, 1, 2. For a more complex dataset like CIFAR-10, we also demonstrates good visual quality, faithfully capturing the essence of each image.

**Principal subspaces of the learned features.** Most generative memory-based methods utilize autoencoders, VAEs, or GANs for replay purposes. The structure or distribution of the learned features $\boldsymbol{Z}_j$ for each class is unclear in the feature space. The features $\boldsymbol{Z}_j$ of the i-CTRL memory, on the other hand, have a clear linear structure. Figure 2 visualizes correlations among all learned features $|\boldsymbol{Z}^\top \boldsymbol{Z}|$, in which we observe clear block-diagonal patterns for both datasets.[4] This indicates the features for different classes $\boldsymbol{Z}_j$ indeed lie on subspaces that are incoherent from one another. Hence, features of each class can be well modeled as a principal subspace in the feature space. A more precise measure of affinity among those subspaces can be found in Appendix D.

**Replay images of samples from principal components.** Since features of each class can be modeled as a principal subspace, we further visualize the individual principal components within each of those subspaces. Figure 4 shows the images replayed from sampled features along the top-4 principal components for different classes, on MNIST and CIFAR-10 respectively. Each row represents samples along one principal component and they clearly show similar visual characteristics but distinctively different from those in other rows. We see that the model remembers different poses of '4' after having learned all remaining classes. For CIFAR-10, the incrementally learned memory remembers representative poses and shapes of horses and ships.

### 4.4 EFFECTIVENESS OF INCREMENTAL REVIEWING

We verify how the incrementally learned LDR memory can be further consolidated with an unsupervised incremental reviewing phase described at the end of Section 3.3. Experiments are conducted on CIFAR-10, with 10 steps.

---

[4]Notice that these patterns closely resemble the similarity matrix of response profiles of object categories from different areas of the inferotemporal cortex, as shown in Extended DataFig.3 of (Bao et al., 2020).

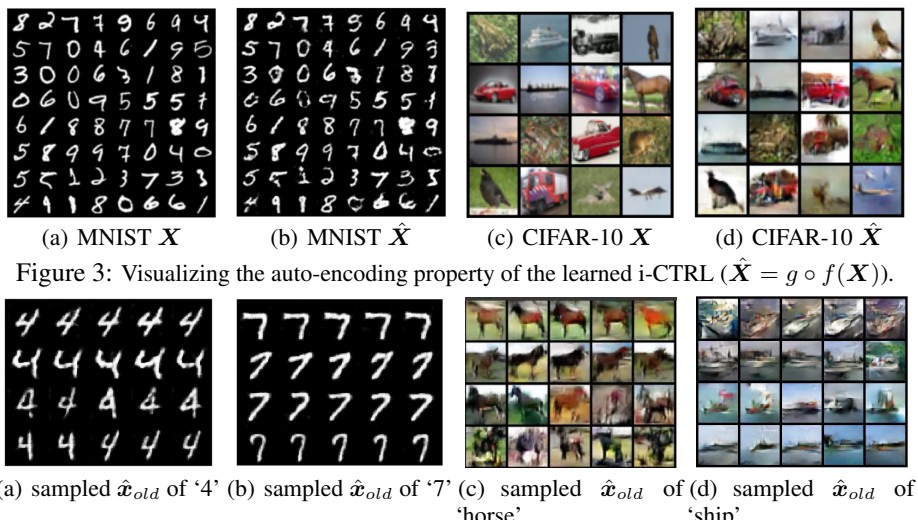

| (a) MNIST $\boldsymbol{X}$ | (b) MNIST $\hat{\boldsymbol{X}}$ | (c) CIFAR-10 $\boldsymbol{X}$ | (d) CIFAR-10 $\hat{\boldsymbol{X}}$ |

Figure 3: Visualizing the auto-encoding property of the learned i-CTRL ($\hat{\boldsymbol{X}} = g \circ f(\boldsymbol{X})$).

(a) sampled $\hat{\boldsymbol{x}}_{old}$ of '4' (b) sampled $\hat{\boldsymbol{x}}_{old}$ of '7' (c) sampled $\hat{\boldsymbol{x}}_{old}$ of 'horse' (d) sampled $\hat{\boldsymbol{x}}_{old}$ of 'ship'

Figure 4: Visualization of 5 reconstructed $\hat{\boldsymbol{x}} = g(\boldsymbol{z})$ from $\boldsymbol{z}$'s with the closest distance to (top-4) principal components of learned features for MNIST (class '4' and class '7') and CIFAR-10 (class 'horse' and 'ship').

**Improving discriminativeness of the memory.** In the reviewing process, all the parameters in the training are the same as incremental learning Table 3 shows how the overall accuracy improves steadily after each cycle of incrementally reviewing the entire dataset. After a few (here 8) cycles, the accuracy approaches the same as that from learning all classes together via Closed-Loop Transcription in a joint fashion (last column). This shows that the reviewing process indeed has the potential to learn a better representation for all classes of data, despite the review process is still trained incrementally.

| # Rev. cycles | 0 | 2 | 4 | 6 | 8 | JL |
|---|---|---|---|---|---|---|
| Accuracy | 0.599 | 0.626 | 0.642 | 0.650 | **0.658** | 0.655 |

Table 3: The overall test accuracies after different numbers of review cycles on CIFAR-10.

**Improving generative quality of the memory.**
Figure 5 left shows replayed images of the first class 'airplane' at the end of incremental learning of all ten classes, sampled along the top-3 principal components – every two rows (16 images) are along one principal direction. Their visual quality remains very decent – observed almost no forgetting. The right figure shows replayed images after reviewing the first class once. We notice a significant improvement in visual quality after the reviewing, and principal components of the features in the subspace start to correspond to distinctively different visual attributes within the same class.

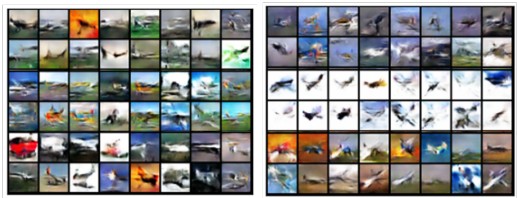

Figure 5: Visualization of replayed images $\hat{\boldsymbol{x}}_{old}$ of class 1-'airplane' in CIFAR-10, before (left) and after (right) one reviewing cycle.

## 5 CONCLUSION

This work provides a simple and unifying framework that can incrementally learn a both discriminative and generative memory for multiple classes of objects. By combining the advantages of a closed-loop transcription system and the simple linear structures of a learned LDR memory, our method outperforms prior work and proves, arguably for the first time, that both stability and plasticity can be achieved *with only a fixed-sized neural system and a single unifying learning objective*. The simplicity of this new framework suggests that its performance, efficiency and scalability can be significantly improved in future extensions. In particular, we believe that this framework can be extended to the fully unsupervised or self-supervised settings, and both its discriminative and generative properties can be further improved.

## ETHICS STATEMENT

All authors agree and will adhere to the conference's Code of Ethics. We do not anticipate any potential ethics issues regarding the research conducted in this work.

## REPRODUCIBILITY STATEMENT

Settings and implementation details of network architectures, optimization methods, and some common hyper-parameters are described in the Appendix B. We will also make our source code available upon request by the reviewers or the area chairs.

## ACKNOWLEDGMENTS AND DISCLOSURE OF FUNDING

Yi Ma acknowledges support from ONR grants N00014-20-1-2002 and N00014-22-1-2102, the joint Simons Foundation-NSF DMS grant #2031899, as well as partial support from Berkeley FHL Vive Center for Enhanced Reality and Berkeley Center for Augmented Cognition, Tsinghua-Berkeley Shenzhen Institute (TBSI) Research Fund, and Berkeley AI Research (BAIR).

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

## A  ALGORITHM OUTLINE

For simplicity of presentation, the main body of this paper has described incremental learning with each incremental task containing one new class of data. In general, however, each incremental task may contain a finite $C$ new classes. In this section, we detail the algorithms associated with i-CTRL in this more general setting.

Suppose we divide the overall task of learning multiple classes of data $\boldsymbol{D}$ into a stream of smaller tasks $\boldsymbol{D}^1, \boldsymbol{D}^2, \ldots, \boldsymbol{D}^t, \ldots, \boldsymbol{D}^T$, where each task consists of labeled data $\boldsymbol{D}^t = \{\boldsymbol{X}^t, \boldsymbol{Y}^t\}$ from $C$ classes, i.e, $\boldsymbol{X}^t = \{\boldsymbol{X}_1^t, \ldots, \boldsymbol{X}_C^t\}$. The overall i-CTRL process is summarized in Algorithm 3

We begin by training the model on the first task $\boldsymbol{D}^1$, optimized via the original objective function (4). We then use FORMING MEMORY MEAN AND COVARIANCE 1 to find $\mathbf{M}^1$, the means and covariances of the representations of classes in the first task. When learning a new task $\boldsymbol{D}^t$, we first sample $\boldsymbol{Z}_{old}$ using MEMORY SAMPLING 2. We then take $(\boldsymbol{X}^t, \boldsymbol{Y}^t)$ from $\boldsymbol{D}^t$, and calculate $\boldsymbol{X}^t \rightarrow \boldsymbol{Z}^t \rightarrow \hat{\boldsymbol{X}}^t \rightarrow \hat{\boldsymbol{Z}}^t$ using $f(\cdot, \theta), g(\cdot, \eta)$ to obtain $\boldsymbol{Z}^t$ and $\hat{\boldsymbol{Z}}^t$. We next compute $\boldsymbol{Z}_{old} \rightarrow \hat{\boldsymbol{X}}_{old} \rightarrow \hat{\boldsymbol{Z}}_{old}$ using $f(\cdot, \theta), g(\cdot, \eta)$. So far, we get $\boldsymbol{Z} = [\boldsymbol{Z}^t, \boldsymbol{Z}_{old}]$ and $\hat{\boldsymbol{Z}} = [\hat{\boldsymbol{Z}}^t, \hat{\boldsymbol{Z}}_{old}]$. The encoder updates $\theta$ by optimizing the objective (8):

$$\max_{\theta} \Delta R(\boldsymbol{Z}) + \Delta R(\hat{\boldsymbol{Z}}) + \lambda \Delta R(\boldsymbol{Z}^t, \hat{\boldsymbol{Z}}^t) - \gamma \Delta R(\boldsymbol{Z}_{old}, \hat{\boldsymbol{Z}}_{old}).$$

The decoder updates $\eta$ via optimizing the objective (9):

$$\min_{\eta} \Delta R(\boldsymbol{Z}) + \Delta R(\hat{\boldsymbol{Z}}) + \lambda \Delta R(\boldsymbol{Z}^t, \hat{\boldsymbol{Z}}^t) + \gamma \Delta R(\boldsymbol{Z}_{old}, \hat{\boldsymbol{Z}}_{old}).$$

We optimize these objectives until the parameters converge. After the training session ends, we calculate $M^t$ of this learn task using FORMING MEMORY MEAN AND COVARIANCE 1 The process of training a new task is repeated until all tasks are learned.

---

**Algorithm 1** FORMING MEMORY MEAN AND COVARIANCE($\boldsymbol{Z}^t$, $k$, $r$)

---

**Require:** $C$ classes of features $\boldsymbol{Z}^t = [\boldsymbol{Z}_1^t, \boldsymbol{Z}_2^t, \ldots, \boldsymbol{Z}_C^t]$ of the entire $t$-th task, $t \in [1, \ldots, T]$. The parameters $k$ and $r$, which correspond to top-$k$ features on each of top-$r$ eigenvectors which we will sample on each class;
 1: **for** $j = 1, 2, \ldots, C$ **do**
 2:     Calculate the top-$r$ eigenvectors $\boldsymbol{V}^j$ of $\boldsymbol{Z}_j^t$. $\boldsymbol{V}^j = [\boldsymbol{v}_1, \ldots, \boldsymbol{v}_r]$ where $\boldsymbol{v}_n$ means the $n$-th eigenvector;
 3:     **for** $i = 1, 2, \ldots, r$ **do**
 4:         Calculate projection distance $\boldsymbol{d} = \boldsymbol{v}_i^\top \boldsymbol{Z}_j^t$;
 5:         Choose the top-$k$ features from $\boldsymbol{Z}_j^t$ based on the distance $\boldsymbol{d}$ to form the set $\boldsymbol{S}_i$;
 6:         Calculate mean $\mu_i$ and covariance $\Sigma_i$ based on the set $\boldsymbol{S}_i$;
 7:     **end for**
 8:     Obtain memory mean and covariance of $j$-th class $\boldsymbol{B}_j \doteq [(\mu_1, \Sigma_1), \ldots, (\mu_r, \Sigma_r)]$;
 9: **end for**
10: Memory of mean and covariance set for $t$-th task $\mathbf{M}^t \doteq [\boldsymbol{B}_1, \ldots, \boldsymbol{B}_C]$.
**Ensure:** $\mathbf{M}^t$

---

## B  IMPLEMENTATION DETAILS

**A simple network architecture.** Tables 4 and 5 give details of the network architecture for the decoder and the encoder networks used for experiments reported in Section 4. All $\alpha$ values in Leaky-ReLU (i.e. lReLU) of the encoder are set to 0.2. We set ($nz = 128$ and $nc = 1$) for MNIST, ($nz = 128$ and $nc = 3$) for CIFAR-10 and CIFAR-100, ($nz = 256$ and $nc = 3$) for ImageNet-50. For ImagetNet-50, we added 2 down sample and up sample layer in f and g respectively to match the resolution of ImageNet-50. The details of architecture are given in Appendix B. The dimension $d$ of the feature space is set accordingly for different datasets, $d =128$ for MNIST and CIFAR-10, $d =256$ for ImageNet-50. More details about the algorithmic settings and ablation studies are given in the Appendix.

---

**Algorithm 2** MEMORY SAMPLING($\mathbf{M}^1, \ldots, \mathbf{M}^t, k, r, C$)

---

**Require:** A set of Memory $\mathbf{M}^1, \ldots \mathbf{M}^t$, where $\mathbf{M}^i \doteq [\boldsymbol{B}_1, \ldots, \boldsymbol{B}_C]$, $k$, $r$ and $C$, which is the number of classes in each task;
Initialize an empty $\boldsymbol{Z}_{old}$;
2: **for** $i = 1, \ldots, t$ **do**
  **for** $j = 1, \ldots, C$ **do**
4:   get $\boldsymbol{B}_j$ from $\mathbf{M}^i$, $\boldsymbol{B}_j \doteq [(\mu_1, \Sigma_1), \ldots, (\mu_r, \Sigma_r)]$;
   For each direction $l \in r$, sample $k$ number of samples from distribution $N(\mu_l, \Sigma_l)$, add them to $\boldsymbol{Z}_{old}$;
6:  **end for**
 **end for**
**Ensure:** $\boldsymbol{Z}_{old}$

---

**Algorithm 3** i-CTRL

---

**Require:** A stream of tasks $\boldsymbol{D}^1, \boldsymbol{D}^2, \ldots, \boldsymbol{D}^T$, where $\boldsymbol{D}^i = \{\boldsymbol{X}^i, \boldsymbol{Y}^i\}$; A pre-trained encoder $f(\cdot, \theta)$ and decoder $g(\cdot, \eta)$ on $\boldsymbol{D}^1$, $k$ and $r$;
Calculate $\boldsymbol{Z}^1$ via $f(\boldsymbol{X}^1, \theta)$;
2: Find $\mathbf{M}^1$ by FORMING MEMORY MEAN AND COVARIANCE($\boldsymbol{Z}^1, k, r$);
 **for** $t = 2, \ldots, T$ **do**
4: Sample $\boldsymbol{Z}_{old}$ = MEMORY SAMPLING($\mathbf{M}^1, \ldots, \mathbf{M}^{t-1}, k, r, C$);
  **while** not converged **do**
6:  Draw samples in $(\boldsymbol{X}^t, \boldsymbol{Y}^t)$ from the $t$-th task $\boldsymbol{D}^t$;
   Compute expressions: $\boldsymbol{X}^t \to \boldsymbol{Z}^t \to \hat{\boldsymbol{X}}^t \to \hat{\boldsymbol{Z}}^t$;
8:  Replay the old memory $\boldsymbol{Z}_{old} \to \hat{\boldsymbol{X}}_{old} \to \hat{\boldsymbol{Z}}_{old}$;
   $\boldsymbol{Z} = [\boldsymbol{Z}^t, \boldsymbol{Z}_{old}]$; $\hat{\boldsymbol{Z}} = [\hat{\boldsymbol{Z}}^t, \hat{\boldsymbol{Z}}_{old}]$;
10:  Update $\theta$ via the optimization objective (8);
   Update $\eta$ via the optimization objective (9);
12:  **end while**
  Calculate $\boldsymbol{Z}^t$ via $f(\boldsymbol{X}^t, \theta)$;
14: Find $\mathbf{M}^t$ by FORMING MEMORY MEAN AND COVARIANCE($\boldsymbol{Z}^t, k, r$);
 **end for**
**Ensure:** $f(\cdot, \theta)$ and $g(\cdot, \eta)$

---

**Optimization settings.** For all experiments, we use Adam (Kingma & Ba, 2014) as our optimizer, with hyperparameters $\beta_1 = 0.5, \beta_2 = 0.999$. Learning rate is set to be 0.0001. We choose $\epsilon^2 = 1.0$, $\gamma = 1$, and $\lambda = 10$ for both equation (8) and (9) in all experiments. For MNIST, CIFAR-10 and CIFAR-100, each task is trained for 120 epochs; For ImageNet-50, the first task $\boldsymbol{D}^1$ is trained for 500 epochs with constraint on augmentation used in (Chen et al., 2020) and 150 epochs for rest incremental 4 tasks using the normal i-CTRL objective 7. All experiments are conducted with 1 or 2 RTX 3090 GPUs.

**Prototype settings** As we use prototype sampling in this method, so the storage becomes almost trivial. For MNIST, we choose $r = 6, k = 10$. For CIFAR-10, we choose $r = 12, k = 20$. For ImageNet-50, we us $r = 10, k = 15$. For CIFAR-100, we us $r = 10, k = 20$.

| $\boldsymbol{z} \in \mathbb{R}^{1 \times 1 \times nz}$ |
|:---:|
| $4 \times 4$, stride=1, pad=0 deconv. BN 256 ReLU |
| $4 \times 4$, stride=2, pad=1 deconv. BN 128 ReLU |
| $4 \times 4$, stride=2, pad=1 deconv. BN 64 ReLU |
| $4 \times 4$, stride=2, pad=1 deconv. 1 Tanh |

Table 4: Network architecture of the decoder $g(\cdot, \eta)$.

| Image $\boldsymbol{x} \in \mathbb{R}^{32 \times 32 \times nc}$ |
|---|
| $4 \times 4$, stride=2, pad=1 conv 64 lReLU |
| $4 \times 4$, stride=2, pad=1 conv. BN 128 lReLU |
| $4 \times 4$, stride=2, pad=1 conv. BN 256 lReLU |
| $4 \times 4$, stride=1, pad=0 conv $nz$ |

Table 5: Network architecture of the encoder $f(\cdot, \theta)$.

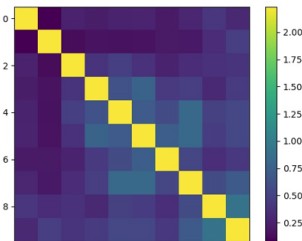

Figure 6: Affinity between memory subspaces within CIFAR-10.

**A simple nearest subspace classifier.** Similar to (Dai et al., 2022) and (Yu et al., 2020), we adopt a very simple *nearest subspace* algorithm to evaluate how discriminative our learned features are for classification. Suppose $\boldsymbol{Z}_j$ are the learned features of the $j$-th class. Let $\boldsymbol{\mu}_j \in \mathbb{R}^d$ be its mean and $\boldsymbol{U}_j \in \mathbb{R}^{d \times r_j}$ be the first $r_j$ principal components for $\boldsymbol{Z}_j$, where $r_j$ is the estimated dimension of class $j$. For a test data $\boldsymbol{x}'$, its feature $\boldsymbol{z}'$ is given by $f(\boldsymbol{x}', \theta)$. Then, its class label can be predicted by $j' = \arg\min_{j \in \{1,\ldots,k\}} \|(\boldsymbol{I} - \boldsymbol{U}_j \boldsymbol{U}_j^\top)(\boldsymbol{z}' - \boldsymbol{\mu}_j)\|_2^2$. It is especially noteworthy that our method does *not* need to train a separate deep neural network for classification whereas most other methods do.

## C   RESOURCE COMPARISON DETAILS

In Table 6 of the appendix, both i-CTRL and EEC methods are tested on CIFAR-10. Under joint learning, CTRL follows the setting in Appendix A.4 of (Dai et al., 2022), but we adopt the architectures of the encoder and decoder detailed in Table 5 and Table 4 respectively. The training batch size is 1600 over 1400 epochs because the generative CTRL model is more challenging to train than the simple classifier network that EEC uses in the joint learning setting; EEC uses the ResNet architecture from (Gulrajani et al., 2017) for the classifier, with training batch size and training epochs of 128 and 100 respectively.

## D   AFFINITY BETWEEN LEARNED SUBSPACES

As we see in Figure 2, the learned features of different classes are highly incoherent and their correlations form a block-diagonal pattern. We here conduct more quantitative analysis of the affinity among subspaces learned for different classes. The analysis is done on features learned for CIFAR-10 using 10 splits with 2000 features. For two subspaces $\mathcal{S}$ and $\mathcal{S}'$ of dimension $d$ and $d'$ , we follow the definition of normalized affinity in (Soltanolkotabi et al., 2014):

$$\text{aff}(\mathcal{S}, \mathcal{S}') \doteq \sqrt{\frac{\sum_i^{d*d'} \cos^2 \theta^i}{d * d'}}. \tag{10}$$

We calculate the $\text{aff}(\mathcal{S}, \mathcal{S}')$ through $\|\boldsymbol{U}^\top \boldsymbol{U}'\|_F$ where $\boldsymbol{U}/\boldsymbol{U}'$ is the normalized column space of features $\boldsymbol{Z}/\boldsymbol{Z}'$ that can be obtained by SVD.

The affinity measures the angle between two subspaces. The larger the value, the smaller the angle. As shown in Figure 6, we see that similar classes have higher affinities. For example, 8-ship and 9-trucks have higher affinity in the figure, whereas 6-frogs has a much lower affinity than these two classes. This suggests that the affinity score of these subspaces captures similarity in visual attributes between different classes.

| Method | Resource | JL | IL | Diff |
|---|---|---|---|---|
| i-CTRL (ours) | Model Size | 2 M | 2 M | same |
| | Train Time | 15 hours | 1.5 hours | 10x faster |
| EEC | Model Size | 1 M | 10 M | 10x larger |
| | Train Time | 0.6 hours | ≥4 hours | 7x slower |

Table 6: The resource comparison on the joint learning (JL) and incremental learning (IL) of different methods. Both methods are tested on CIFAR-10 and the details of the comparison setting can be found in Appendix C.

# E    INCREMENTAL LEARNING VERSUS JOINT LEARNING.

One main benefit of incremental learning is to learn one class (or one small task) at a time. So it should result in less storage and computation than jointly learning. Table 6 shows this is indeed the case for our method: IL on CIFAR-10 is 10 times faster than JL.[5] However, this is often not the case for many existing incremental methods such as EEC (Ayub & Wagner, 2020), the current SOTA in generative memory-based methods. Not only does its incremental mode require a much larger model size (than its joint mode and ours[6]), it also takes significantly (7 times) longer to train.

# F    ABLATION STUDIES

We conduct all ablation studies under the setting of CIFAR-10 split into 5 tasks with feature size of 2000, and default values of $k = 20$, $r = 12$, $\lambda = 10$, and $\gamma = 1$. We use the average incremental accuracy as a measure for these studies.

## F.1    IMPACT OF CHOICE OF OPTIMIZATION PARAMETERS

**Parameter $m$ and $r$ for memory sampling.** Here, we verify the impact of the memory size of Algorithm 1 on the performance of our method. The feature size is determined by two hyperparameters $r$, which is the number of the PCA directions and $m$, which is the number of sampled features around each principal direction. The value of $r$ varies from 10 to 14, and the value of $m$ varies from 20 to 40. Table 7 reports the results of the average incremental accuracy. From the table, we observe that as long as the selection of $m$ and $r$ are in a reasonable range, the overall performance is stable.

| | $m$=20 | $m$=30 | $m$=40 |
|---|---|---|---|
| $r$=10 | 0.713 | 0.720 | 0.728 |
| $r$=12 | 0.719 | 0.727 | 0.725 |
| $r$=14 | 0.718 | 0.721 | 0.725 |

Table 7: Ablation study on varying $m$ and $r$ in PROTOTYPESAMPLING, in terms of the average incremental accuracy.

**Hyperparameter $\lambda$ and $\gamma$ in the learning objective.** $\lambda$ and $\gamma$ are two important hyperparameters in the objective functions for both (8) and (9). Here, we want to justify our selection of $\lambda$ and $\gamma$ and demonstrate the stability of our method to their choices. We analyze the sensitivity of the performance to the $\lambda$ and $\gamma$ respectively. In Table 8, we set $\gamma = 1$ and change the value of $\lambda$ from 0.1 to 50. The results indicate the accuracy becomes low only when $\lambda$ are chosen to be extreme (e.g 0.1, 1, 50). We then change the value of $\gamma$ in a large range from 0.01 to 100 with $\lambda$ fixed at 10. Results in Table 9 indicate that the accuracy starts to drop when $\gamma$ is larger than 10. Hence, in all our experiments reported in Section 4, we set $\lambda = 10$ and $\gamma = 1$ for simplicity.

---

[5]Note in our method, both JL and IL optimize on the same network. The JL mode is trained on all ten classes together, hence it normally takes more epochs to converge and longer time to train. But the IL mode converges much faster, as it should have.

[6]For EEC, since its classifier and generators are separated, under the JL setting, it only needs a 8-layers convolutional network to train a classifier for all classes. In the incremental mode, it requires multiple generative models. Note that our JL model is also a generative model hence requires more time to train as well.

| $\lambda$ | 0.1 | 1 | 10 | 20 | 50 |
|---|---|---|---|---|---|
| Accuracy | 0.592 | 0.620 | 0.712 | 0.701 | 0.691 |

Table 8: Ablation study on varying $\lambda$ in terms of the average incremental accuracy.

| $\gamma$ | 0.01 | 0.1 | 1 | 10 | 100 |
|---|---|---|---|---|---|
| Accuracy | 0.713 | 0.716 | 0.712 | 0.700 | 0.655 |

Table 9: Ablation study on varying $\gamma$ in terms of the average incremental accuracy.

## F.2 SENSITIVITY TO CHOICE OF RANDOM SEED

It is known that some incremental learning methods such as (Kirkpatrick et al., 2017) can be sensitive to random seeds. We report in Table 10 the average incremental accuracy of i-CTRL with different random seeds (conducted on CIFAR-10 split into 10 tasks, with a feature size 2000). As we can see, the choice of random seed has very little effect on the performance of i-CTRL.

| Random Seed | 1 | 5 | 10 | 15 | 100 |
|---|---|---|---|---|---|
| Average Accuracy | 0.720 | 0.720 | 0.720 | 0.720 | 0.721 |
| Last Accuracy | 0.594 | 0.592 | 0.593 | 0.594 | 0.594 |

Table 10: Ablation study on varying random seeds.

## F.3 THE SIGNIFICANCE OF AUGMENTATION IN TRAINING IMAGENET-50

In this section, we study the the impact of using augmentation from Chen et al. (2020) to train the first task has on our method. From Tab 11, we conclude that augmentation did help the model the learn better representation. Even without it, it has shown that our method still outperform the current generative-replay based method by more than 10%. Through this experiment, we think that add augmentation may be the solution for generative-replay based methods to scale up to even larger datasets. We leave that to future study.

## F.4 THE SIGNIFICANCE OF CONSTRAINT IN MINMAX OPTIMIZATION

Here, we want to justify the significance of this constraint in the context of incremental learning. We report in table 12 the performance of i-CTRL with and without constraint. Without the constraint, i-CTRL fall into the victim of catastrophic forgetting. We can conclude that constraint has played a significant role in the success of our method.

## G COMPARISON WITH MORE BASELINES

Due the limitation of space in main paragraph, we present here a table with more comparison with other methods.

From the table, we see that comparing to the previous methods especially exemplar-based methods, our method still leads them numerically. We have also conducted on experiments on CIFAR-100. On more complex data such as CIFAR-100(Krizhevsky et al., 2014), it is also observed in Tab 14 that i-CTRL has led the current exemplar-based methods. It is noteworthy that there is no generative-replayed based methods in the table. Since it hard for many of the current generative-replay based methods to scale up to more complex setting.

## H QUANTITATIVE EVALUATION OF LEARNED GENERATOR

In this section, we use FID score (Heusel et al., 2017) and Inception Scores (IS) (Salimans et al., 2016) to quantitatively measure the performance of our incrementally learned generator. As there

| iCaRL-S | EEIL-S | DGMw | EEC | EECS | i-CTRL(without aug) | i-CTRL(with aug) |
|---------|--------|------|-----|------|---------------------|------------------|
| 0.290 | 0.118 | 0.178 | 0.352 | 0.309 | **0.458** | **0.523** |

Table 11: Comparison on ImageNet-50. The results of other methods are as reported in the EEC paper.

|  | With Constraint | Without Constraint |
|--|-----------------|--------------------|
| Average Accuracy | 0.723 | 0.380 |
| Last Accuracy | 0.627 | 0.223 |

Table 12: Ablation study on the important of constraint in (7)

| Method | MNIST | | | | CIFAR-10 | | | |
|--------|-------|-----|-----|-----|----------|-----|-----|-----|
|  | 10-splits | | 5-splits | | 10-splits | | 5-splits | |
|  | Last | Avg | Last | Avg | Last | Avg | Last | Avg |
| *Regularization* | | | | | | | | |
| LwF (Li & Hoiem, 2017) | - | - | 0.196 | 0.455 | - | - | 0.196 | 0.440 |
| SI (Zenke et al., 2017) | - | - | 0.193 | 0.461 | - | - | 0.196 | 0.441 |
| *Architecture* | | | | | | | | |
| ReduNet (Wu et al., 2021) | - | - | 0.961 | 0.982 | - | - | 0.539 | 0.645 |
| *Exemplar* | | | | | | | | |
| iCaRL (Rebuffi et al., 2017) | 0.322 | 0.588 | 0.725 | 0.803 | 0.212 | 0.431 | 0.487 | 0.632 |
| A-GEM (Chaudhry et al., 2018) | 0.382 | 0.574 | 0.597 | 0.764 | 0.115 | 0.293 | 0.204 | 0.473 |
| CLR-ER (Arani et al., 2022) | - | - | - | 0.895 | - | - | - | 0.662 |
| ER-Reservoir (Chaudhry et al., 2019b) | - | - | - | - | - | - | - | 0.685 |
| GDumb (Prabhu et al., 2020) | - | - | - | 0.919 | - | - | - | 0.618 |
| Rainbow Memory (Bang et al., 2021) | - | - | 0.927 | - | - | - | - | - |
| DER++ (Buzzega et al., 2020) | - | - | - | - | - | - | - | 0.648 |
| *Generative Memory* | | | | | | | | |
| DGMw (Ostapenko et al., 2019) | - | 0.965 | - | - | - | 0.562 | - | - |
| EEC (Ayub & Wagner, 2020) | - | 0.978 | - | - | - | 0.669 | - | - |
| EECS (Ayub & Wagner, 2020) | - | 0.963 | - | - | - | 0.619 | - | - |
| i-CTRL (ours) | **0.975** | **0.989** | **0.978** | **0.990** | **0.599** | **0.727** | **0.627** | **0.723** |

Table 13: Comparison on MNIST and CIFAR-10.

| Method | Last Accuracy 5-split | Last Accuracy 10-split | Last Accuracy 20-split |
|--------|----------------------|------------------------|------------------------|
| LwF(Li & Hoiem, 2017) | - | 0.252 | 0.141 |
| iCaRL(Rebuffi et al., 2017) | - | 0.346 | - |
| GDUMB(Prabhu et al., 2020) | - | - | 0.241 |
| Rainbow Memory(Bang et al., 2021) | 0.414 | - | - |
| i-CTRL | 0.435 | 0.392 | 0.378 |

Table 14: Comparison on CIFAR-100.

|  | IS↑ | FID↓ |
|--|-----|------|
| DCGAN (Radford et al., 2016) | 6.6 | 37.4 |
| i-CTRL (ours) | 6.5 | 36.7 |

Table 15: Comparison IS and FID on CIFAR-10

exist very few generative-based or replay-based incremental methods offer a quantitative result for us to compare. We here compare with DCGAN (Radford et al., 2016), which is the backbone of our method, trained jointly for all classes. Based on table15, it is seen that our method has competitive FID and IS score comparing to DCGAN. Hence, despite trained incrementally, our method still generates high-quality images.

# I  i-CTRL IN EXTREME SETTING

In this section, we conduct some ablation study of i-CTRL implemented in extreme settings.

## I.1  IMBALANCED DATASETS

Often in real life, the data we encounter are not perfectly balanced. To testify our model's performance in this situation, we conduct experiment on imbalance-CIFAR-10. In this subsection, CIFAR-10 is split into 5 tasks, with task 2 and task 4 having only half of the original data. We call this setting imbalance-CIFAR-10. i-CTRL is trained with parameters same as section B. From table 16, we observe that imbalance CIFAR-10 has very little impact on the performance of our method.

| | CIFAR-10 (balance) | | CIFAR-10 (imbalance) | |
|---|---|---|---|---|
| | Last Accuracy | Average Accuracy | Last Accuracy | Average Accuracy |
| i-CTRL | 0.627 | 0.727 | 0.623 | 0.723 |

Table 16: Comparison of i-CTRL performance on CIFAR-10 and imbalance-CIFAR-10

## I.2  SMALL SUBSET OF DATA

Another interesting extreme scenario to examine would be small subset of dataset. Again, we may not get large number of dataset for us to train every time. To testify i-CTRL's performance in this scenario, we design small-CIFAR-10. For example, We denote CIFAR-10(50%), meaning we have deleted 50% of data from every class in CIFAR-10. We run i-CTRL on CIFAR-10(20%), CIFAR-10(40%), CIFAR-10(60%), CIFAR-10(80%), CIFAR-10(100%) *without* tuning any parameter. From Table 17, we observe that smaller daatset will have impact on the performance of our method. If the portion is larger than 20%, the impact is relatively small. When the size of data reduces to only 20%, the impact becomes larger. Nonetheless, since CIFAR-10(20%) is nearly a new dataset, we can reduce the impact by tuning parameters. In Table 18, we present results of i-CTRLon CIFAR-10(20%) after

| | CIFAR-10 (balance) | |
|---|---|---|
| | Last Accuracy | Average Accuracy |
| CIFAR-10(20%) | 0.476 | 0.625 |
| CIFAR-10(40%) | 0.575 | 0.691 |
| CIFAR-10(60%) | 0.589 | 0.709 |
| CIFAR-10(80%) | 0.615 | 0.721 |
| CIFAR-10(100%) | 0.627 | 0.727 |

Table 17: Comparison of i-CTRL performance on different scales of CIFAR-10

tuning the hyperparamter $\lambda$ and epochs. Since CIFAR-10(20%) is a very small dataset, we reduce the number of $\lambda$ and epochs to avoid forgetting previous learned classes. We observe that smaller $\lambda$ and epochs can greatly improve the performance of i-CTRL on very small subset of data like CIFAR-10(20%).

| | CIFAR-10 (balance) | |
|---|---|---|
| | Last Accuracy | Average Accuracy |
| CIFAR-10(20%), $\lambda = 10$, epochs=120 | 0.476 | 0.625 |
| CIFAR-10(20%), $\lambda = 5$, epochs=60 | 0.541 | 0.671 |

Table 18: Comparison of i-CTRL performance on CIFAR-10(20%) with different hyperparameters

# J USING AFFINITY TO MEASURE THE PERFORMANCE

In this section, we discuss if affinity between the memory subspaced learned can be used to evaluate the performance of our method. Following the setting of subset in CIFAR-10, we visualize the affinity in Fig 7. From the figure, we see that as the subset of CIFAR-10 becomes smaller, the affinity learned by i-CTRL becomes more distant. It can be used as a sign for unsatisdying performance because ideally, we would want the affinity between similar classes (truck and car) to be close. If the affinity graph shows that the model does not capture this kind of relationship, it is a sign that the overall performance could be worse.

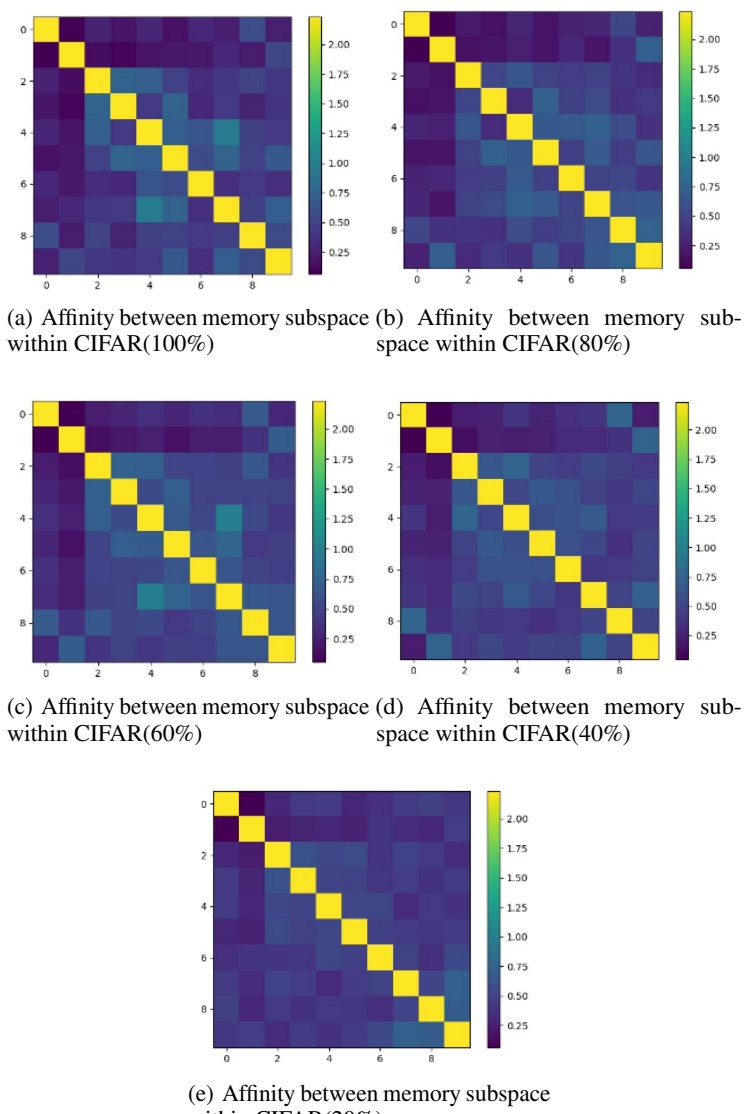

(a) Affinity between memory subspace within CIFAR(100%)

(b) Affinity between memory subspace within CIFAR(80%)

(c) Affinity between memory subspace within CIFAR(60%)

(d) Affinity between memory subspace within CIFAR(40%)

(e) Affinity between memory subspace within CIFAR(20%)

Figure 7: Visualization of the affinity between memory subspace under different subset of CIFAR-10

