# OpenReview forum: "Incremental Learning of Structured Memory via Closed-Loop Transcription"
_ICLR.cc/2023/Conference — ICLR 2023 poster_

### Official Review · Reviewer_HArM · 2022-10-22

**Confidence:** 4
**Correctness:** 4
**Technical Novelty And Significance:** 3
**Empirical Novelty And Significance:** 4
**Recommendation:** 8

**Clarity, Quality, Novelty And Reproducibility:**

The paper is clearly written. It is explicit about the techniques it has adopted from other papers, and the use of them appears to be novel, though it would be helpful to include a clearer description of, for example, how the use of the rate reduction objective is different in (Wu et al 2021), and in what setting the closed loop transcription is used in Dai et al 2022. Algorithmic pseudo-code is provided in the appendix but no actual code is provided in the supplementary material.

**Strength And Weaknesses:**

Strengths
- The method demonstrates impressively strong results on a number of tasks in the challenging incremental learning setting, improving greatly over relevant baselines.
- The idea is compelling, and although it uses techniques from previous works (Wu et al 2021, Dai et al 2022), it is adapted uniquely for the incremental class learning setting restricted to a fixed network capacity. The novelty of the method, its minimal memory footprint and the impressive results will certainly be of interest to the continual learning community.
- The paper is easy to read and the method clearly described.

Areas for improvement / questions
- Though it is clear that I-CTRL has the advantage over Redunet that (i) the results are better and (ii) it does not require dynamic expansion of the network, since they seem to use the same objective function, it would be good to have a clearer description of how the two methods differ in how they use it.
- Minimax objectives can often be unstable and sensitive to hyperparameters - was there evidence of instability during training and how sensitive is the performance to the values of lambda and gamma that weight the rate reduction losses for the new features and old features respectively?
- Though the results without augmentation are reported in the appendix are still better than the baselines ( though 6-7% worse accuracy than with augmentations), would it not be fairer to use these as comparison to other methods, which presumably do not use augmentation (from a glance at the EEC paper, it does not).
- Many would disagree with the decision to discard methods that keep raw exemplars as unfair comparisons and on this basis call the results of I-CTRL SOTA (which they would not be otherwise, e.g. Dark Experience Replay outperforms I-CTRL on CIFAR-10 with a relatively modest buffer size of 500 exemplars). Indeed ICTRL still makes use of a memory buffer and one version of iCTRL that is advertised involves a form of replay of previous exemplars. While I certainly believe that the minimal memory capacity required by I-CTRL is a great feature and is rightly highlighted in the paper, the SOTA claim is perhaps a little strong given that one can easily argue that replay-based methods are very practical continual learning methods given that memory storage is generally cheap.
- It is mentioned in the introduction that EWC requires application-specific external mechanisms - what aspect of EWC is application-specific?

**Summary Of The Paper:**

This paper proposes a method, I-CTRL, for incremental class learning that learns a ‘closed loop transcription’ between each of the classes and a corresponding subspace of a low-dimensional feature space, know as a linear discriminative representation. The closed loop transcription is achieved by optimising an autoencoder (that maps the inputs to the feature space and back) with a rate reduction-based minimax objective. The objective ensures that the subspaces are highly incoherent to each other, such that updating one does not interfere with the others, preventing catastrophic forgetting. Furthermore, since the subspaces are linear, each class can be summarised with an estimated mean and covariance matrix in the feature space. These summaries are stored in memory as training progresses and are then used to optimise the minimax objective in feature space by essentially ensuring that the features are consistent with their sampled reconstructions through the closed loop transcription. Experiments are run on a number of incremental class learning benchmarks (MNIST, CIFAR10 and Imagenet-50), where the method is shown to outperform all the baselines, which include the SOTA among methods that (a) do not expand their architecture throughout training or (b) maintain a large number of raw exemplars. It is also shown that the method can benefit from unsupervised replay of old examples. The method can be used as both a discriminative and a generative model, and some visualisations are provided that demonstrate the quality of generated samples, as well as the block-diagonal structure of the learnt feature space.

**Summary Of The Review:**

This paper presents a novel approach to continual learning along with impressive results on incremental class learning benchmarks with a minimal memory footprint. While the claims of SOTA are potentially misleading, when a large class of methods (dynamic architectures and replay-based methods) are questionably discarded as unfair comparisons, the results are undoubtedly strong and will be of great interest to the CL community.

---

> ### Author Response · Authors · 2022-11-17
> **Response to Reviewer HArM**
>
> We thank the reviewer for the valuable comments and suggestions! For the comment you have, we have provided some clarifications to answer them. We have also updated the paper based on your comments.
>
> >Q. Though it is clear that I-CTRL has the advantage over Redunet that (i) the results are better and (ii) it does not require dynamic expansion of the network, since they seem to use the same objective function, it would be good to have a clearer description of how the two methods differ in how they use it.
>
> A: Thank you for the comment! We have updated the related works section to further clarify the differences.
>
> > Q. Minimax objectives can often be unstable and sensitive to hyperparameters - was there evidence of instability during training and how sensitive is the performance to the values of lambda and gamma that weight the rate reduction losses for the new features and old features respectively?
>
> A: Thank you for the comment. Please refer to Table 8 and 9 of Section F.1 of Appendix, where we observe a high tolerance toward the choice of lambda and gamma.
>
> > Q. Though the results without augmentation are reported in the appendix are still better than the baselines ( though 6-7% worse accuracy than with augmentations), would it not be fairer to use these as comparison to other methods, which presumably do not use augmentation (from a glance at the EEC paper, it does not).
>
> A: We thank you for the reviewer’s comment. Following your advice, we have used the without-augmentation result as our baseline in the comparison. Instead, we leave the results with augmentation in the appendix as some further discussion on how to further improve the results.
>
> > Q. Many would disagree with the decision to discard methods that keep raw exemplars as unfair comparisons and on this basis call the results of I-CTRL SOTA (which they would not be otherwise, e.g. Dark Experience Replay outperforms I-CTRL on CIFAR-10 with a relatively modest buffer size of 500 exemplars). Indeed ICTRL still makes use of a memory buffer and one version of iCTRL that is advertised involves a form of replay of previous exemplars. While I certainly believe that the minimal memory capacity required by I-CTRL is a great feature and is rightly highlighted in the paper, the SOTA claim is perhaps a little strong given that one can easily argue that replay-based methods are very practical continual learning methods given that memory storage is generally cheap.
>
> A:  We thank the reviewer for this kind suggestion! We have reworded the sections related to this more carefully following your suggestion.
>
> > Q. It is mentioned in the introduction that EWC requires application-specific external mechanisms - what aspect of EWC is application-specific?
>
> A: Thank you for pointing this out! We have reworded this phrase to “require external task information [EWC]”.
>
> We hope the above answers can clarify your questions. We would like to thank you again for the insightful questions and recognition of our work!

---

### Official Review · Reviewer_72tt · 2022-10-23

**Confidence:** 4
**Correctness:** 4
**Technical Novelty And Significance:** 2
**Empirical Novelty And Significance:** 3
**Recommendation:** 8

**Clarity, Quality, Novelty And Reproducibility:**

The authors present their motivations and contributions in a precise way. Despite only adding minor modifications to what was presented in past works, these are important for the correct functioning of the method.

I believe that using generative models is a line of research with a lot of potentials.

**Strength And Weaknesses:**

S:
- The idea presented is very well-motivated not only from CL's point of view but also from the perspective of how we accumulate knowledge as humans.
- The fact that it is a generative model with fixed capacity makes it an exciting proposal since it does not require adding new weights to a model or re-training models with the problems of overfitting a small group of data.
- Applying CTRL to an incremental learning environment is a smart idea. Accumulating "classes" in separate subspaces and saving this space's mean and covariance can help to mitigate forgetting.
- In general, the ideas are well explained, and the structure of the paper is adequate.
- The experiments carried out in section 4.3 are interesting. It might have been interesting to see how it behaves with larger images like ImageNet.

W:
- One problem is that the method is incremental from CTRL. The authors propose only minor (but critical) modifications to CTRL to apply it to CL.
- A possible limitation of this model is the subspaces that the model finds. I imagine that as the number of classes increases, finding incoherent subspaces becomes more challenging. To the authors: Did you test with benchmarks with more classes? Like CIFAR100 or TinyImageNet, which has 200 classes.

Q:
- I wonder if the first subspaces that the model learns can affect the model's behavior in the future. I imagine that learning easy or difficult subspaces in the first tasks can affect in one way or another the subspaces that the model finds later. What are your opinions about it?
- The constraint added in Eq 8 and 9 can be seen as a regularization similar to distillation on the covariance. The gamma value regulates the importance of maintaining the subspace of the previous classes. Did you carry out experiments on how this value affects training?
- What does the Union in Eq3 represent? The concatenation?
- How many times was each experiment run? Could you get the std from the experiments?
- I like the idea of ​​using the nearest neighbor as a classifier. The other methods used a typical classifier or something like iCARL? There is evidence that most forgetting occurs in the classifier, so it might be interesting to compare it equally.
- In the section "Improving discriminativeness of memory". The factor that regulates the difference between covariances, is used? Or is using gamma equal to zero?

I recommend changing Fig5. It is not easy to see that the photos are grouped in rows of two.

**Summary Of The Paper:**

In this work, the authors present a method to mitigate forgetting in Continual Learning based on closed-loop transcription (CTRL). The authors adapt CTRL to condense the classes' information in a memory composed of an encoder and decoder capable of generating representations of previous tasks. Using linear discriminative representations, CTRL encourages the models to learn compressed representations corresponding to different classes so as to maximize discriminative and generative properties. Taking advantage of these features, the authors propose to store the mean and covariance of the Z representations for each class, and then use these to mitigate forgetting, aiming for the covariance to remain stable while training a new task. The authors show promising results in various benchmarks.

**Summary Of The Review:**

I found the paper well written, with its motivations and results correctly showing the contributions. My only concern is a possible limitation not mentioned by the authors: the capacity to store many tasks/classes of the subspace (what happens when we add more than 50 tasks).

---

> ### Author Response · Authors · 2022-11-17
> **Response to Reviewer 72tt**
>
> We thank the reviewer for the valuable comments and suggestions! For the comment you have, we have provided some clarification and additional experiments to answer them. We have also updated the paper based on your comments.
>
>
> >Q: The constraint added in Eq 8 and 9 can be seen as a regularization similar to distillation on the covariance. The gamma value regulates the importance of maintaining the subspace of the previous classes. Did you carry out experiments on how this value affects training?
>
> A: Thank you for pointing it out! Yes, we did ablation study on the choice of lambda and gamma. Please refer to Table.8 and Table 9 of the Appendix F.1 to see the details. From the experiment, we observe that our method has a high tolerance on the choice of lambda and gamma.
>
> >Q: What does the Union in Eq3 represent? The concatenation?
>
> A: Thank you for pointing it out. Yes, you can understand it as concatenation. We have added clarification in the main text
>
> > Q: I wonder if the first subspaces that the model learns can affect the model's behavior in the future. I imagine that learning easy or difficult subspaces in the first tasks can affect in one way or another the subspaces that the model finds later. What are your opinions about it?
>
> A: Yes, we agree that  the learned subspace of the first task is crucial in our method. We think this can be reflected in the experiment about ImageNet-50 in Appendix F.3 where we study the effect of augmentation in the first task. With the help of augmentation in the first task, the model learns a better subspace at first. And, it has shown that it helps the overall performance. Additionally, we think that incremental reviewing we introduced in section 3 can help to further learn a good subspace.
>
> > Q. A possible limitation of this model is the subspaces that the model finds. I imagine that as the number of classes increases, finding incoherent subspaces becomes more challenging. To the authors: Did you test with benchmarks with more classes? Like CIFAR100 or TinyImageNet, which has 200 classes.
>
> A: We agree that this is a potential limitation of our work, and have added more discussion about it in the main paragraph. While we cannot currently provide more extensive results given the time constraints of the rebuttal period, we have  runned  our method on CIFAR-100 equally split into 5 tasks following the practice of other literature. Detailed implementation and table of comparison can be found in Appendix G. Unfortunately, we didn’t find other generative-based methods that scale up to 100 classes for comparison, so we choose replay-based methods with more storage as a comparison. From the experiments, we observe that our method is able to achieve results on par with replay-based methods under slightly unfair comparison.
>
> >Q: How many times was each experiment run? Could you get the std from the experiments?
>
> A: Thank you for your comments. Please refer to the F.2 of Appendix, we have run ablation studies on the sensitivity of our method to the random seed. Our method shows good robustness to random seed. The standard deviation of random seed is around 0.5%.
>
> >Q: I like the idea of ​​using the nearest neighbor as a classifier. The other methods used a typical classifier or something like iCARL? There is evidence that most forgetting occurs in the classifier, so it might be interesting to compare it equally.
>
> A:  This is an interesting point to raise; as the goal of our work is to continually learn linear discriminative representation for incoming classes, we adopt the nearest subspace classifier, which was suggested as the optimal classifier for linear discriminative representation methods. For methods that follow iCARL that use the nearest average representation to classify, we think our classifier shares a similar idea to that line of research. For line of work using a typical classifier, we have included more discussion in the paper on it following your advice. In general, we think our work aims to learn a continual representation whereas this line of work aims to learn a continual classifier. We thank the reviewer for raising this point.
>
> >Q: In the section "Improving discriminativeness of memory". The factor that regulates the difference between covariances, is used? Or is using gamma equal to zero?
>
> A: Thank you for pointing this out! We have added more clarification in that section. To answer the question, gamma is used in this section as usual. We consider it be a contribution of this work that improving discriminativeness of memory can be trained without changing the hyperparameters.
>
> >Q: I recommend changing Fig5. It is not easy to see that the photos are grouped in rows of two.
>
> A: We thank you for the reviewer’s comment. We fix it in the revised version.
>
> We hope the above answers can clarify your questions. Thank you again for the insightful questions and recognition of our work!

---

> > ### Comment · Reviewer_72tt · 2022-11-20
> > **Thanks for the Response**
> >
> > I appreciate the response from the authors. These answers helped alleviate some of my doubts about the method and the document. Even though I think the paper can still be improved (better initial space analysis (first task) and better comparisons with other methods in environments with more tasks), I believe it is a paper that can be accepted at the conference as it stands. For this reason, I am improving the score.

---

### Official Review · Reviewer_sS4d · 2022-10-26

**Confidence:** 4
**Correctness:** 4
**Technical Novelty And Significance:** 4
**Empirical Novelty And Significance:** 4
**Recommendation:** 8

**Clarity, Quality, Novelty And Reproducibility:**

- Well written
- Quality: High
- Reproducibility -- Not discussed in the paper

**Strength And Weaknesses:**

Strength
- Paper has extensive theoretical and empirical analysis
- It is written coherently and studies related work carefully

**Summary Of The Paper:**

The paper provides a framework for incrementally learning a generative and discriminative model for multi-class problems using a Linear Discriminative Representation (LDR) and a coding rate reduction objective. The closed-loop transcription framework (CTRL) is formulated as a minimax game wherein the encoder tries to maximize the rate reduction objective and the decoder minimizes it instead. The "distance" between Gaussian ensemble subspaces is estimated.


**Summary Of The Review:**

The paper provides a framework for incrementally learning a generative and discriminative model for multi-class problems using a Linear Discriminative Representation (LDR) and a coding rate reduction objective.

The closed-loop transcription framework (CTRL) is formulated as a minimax game wherein the encoder tries to maximize the rate reduction objective and the decoder minimizes it instead. The "distance" between Gaussian ensemble subspaces is estimated.

The authors state "To this end, we simply sample m representative prototype features on the subspace along its top r principal components, and denote these features as Zj,old" -> This step clearly depends on how well the principal components capture the variance of the data in the first place and affects the samples collected using the mean and variance of Zj,old.
a. How robust is the scheme with imbalanced classes?
b. If incremental learning process learns from a small subset of data, this could affect the principal components (and hence their variances). How is this dealt with?
c. Are there empirical results to justify how a change in percentage of variance of principal components affects the overall performance of the algorithm -- in terms of performance metrics, imbalanced multiple classes and relevant details?
d. Can affinity based subspaces be used to measure the effect of percentage of variance captured by principal components?

Appendix A contains a lot of crucial algorithmic details and the authors should consider incorporating those in the main paper.

---

> ### Author Response · Authors · 2022-11-17
> **Response to Reviewer sS4d**
>
> We thank the reviewer for the valuable comments and suggestions! For the comment you have, we have provided some clarification and additional experiments to answer them. We have also updated the paper based on your comments.
>
> > Q1. How robust is the scheme with imbalanced classes?
>
> A: We thank the reviewer for bringing up this very interesting case! We have conducted additional experiments on imbalance-CIFAR-10, in which we deleted half of the data in class 2,3,6,7.  Please refer to Appendix I.1 to see the details.  In general, we found that our method is robust to an imbalanced dataset. Please let us know if you have any further questions on it.
>
> > Q2.If incremental learning process learns from a small subset of data, this could affect the principal components (and hence their variances). How is this dealt with?
>
> A: This is another very interesting question! We have also conducted additional experiments on this. Using CIFAR-10 as an example, we shrink the dataset to CIFAR-10(40%) such that each class only contains 40% of the original data. We conducted experiments on CIFAR-10(20%), CIFAR-10(40%), CIFAR-10(60%), CIFAR-10(80%), CIFAR-10(100%) with the same hyperparameters. Empirically, we found that when the portion of data is greater than 20%, the impact on our method is relatively small.  When the reduction is as large as 20%, we found that tuning the hyperparameter can effectively improve the performance of i-CTRL. Please refer to Appendix I.2 to see the details
>
>
> > Q3. Can affinity based subspaces be used to measure the effect of percentage of variance captured by principal components?
>
> A: Yes, we believe the affinity would be a good metric to measure the captured variance by principal components. We have included additional results in Appendix J, where we plot the affinity between subspaces learned from various subsets of the training data. When more data is used, the affinity indicates that more structure is captured by principal components. Hence, it could be a good measurement for the subspace learned.
>
> We hope the above answers can clarify your questions. We would like to thank you again for the insightful questions and recognition of our work!

---

### Decision · Program_Chairs · 2023-01-20

**Decision:**

Accept: poster

**Justification For Why Not Higher Score:**

Due to the fact that a previously proposed idea, the CTRL theory, plays the major role in the model. In other words, it may be seen to be simple to apply CTRL for continual learning.

**Justification For Why Not Lower Score:**

The good performance, simplicity, and efficiency of the proposed method indicates that this is a good method to share with the continual learning community.

**Metareview: Summary, Strengths And Weaknesses:**

This paper proposes a method for class-incremental continual learning for generative and discriminative models. The proposed method is based on closed-loop transcription (CTRL) theory to learn linear discriminative representations and apply it to continual learning. The proposed method is simple and its performance and efficiency is well demonstrated in the experiments on MNIST, CIFAR-10, and ImageNet-50.

Strength. The paper is well written. The benefit of CTRL theory for incremental learning is well motivated and studied. Most importantly, it performs well but still quite simple and efficient. The method requires fixed capacity (i.e., it does not require adding new weights) for increasing classes.

Weakness. One may see it somewhat incremental as it takes CTRL to apply to continual learning. But, I think that the conception that CTRL can be beneficial for continual learning itself is a contribution. The minimax game objective can be unstable to train. But in Appendix and rebuttal, the authors show that it is quite robust to different choices of hyper-parameters.


**Note From Pc:**

if the above contains the word "oral" or "spotlight" please see: "oral" presentation means -> notable-top-5% and "spotlight" means -> notable-top-25%. As stated in our emails, we are disassociating presentation type from AC recommendations